# AEG-1 Regulates TWIK-1 Expression as an RNA-Binding Protein in Astrocytes

**DOI:** 10.3390/brainsci11010085

**Published:** 2021-01-11

**Authors:** Hyun-Gug Jung, Ajung Kim, Seung-Chan Kim, Jae-Yong Park, Eun Mi Hwang

**Affiliations:** 1Brain Science Institute, Korea Institute of Science and Technology (KIST), Seoul 02792, Korea; hyungug.jung@einsteinmed.org (H.-G.J.); kitkim819@kist.re.kr (A.K.); 216008@kist.re.kr (S.-C.K.); 2School of Biosystem and Biomedical Science, College of Health Science, Korea University, Seoul 02841, Korea; 3Division of Bio-Medical Science & Technology, KIST School, Korea University of Science and Technology, Seoul 02792, Korea

**Keywords:** astrocyte, AEG-1, TWIK-1, RNA-binding protein

## Abstract

AEG-1, also called MTDH, has oncogenic potential in numerous cancers and is considered a multifunctional modulator because of its involvement in developmental processes and inflammatory and degenerative brain diseases. However, the role of AEG-1 in astrocytes remains unknown. This study aimed to investigate proteins directly regulated by AEG-1 by analyzing their RNA expression patterns in astrocytes transfected with scramble shRNA and AEG-1 shRNA. AEG-1 knockdown down-regulated TWIK-1 mRNA. Real-time quantitative PCR (qPCR) and immunocytochemistry assays confirmed that AEG-1 modulates TWIK-1 mRNA and protein expression. Electrophysiological experiments further revealed that AEG-1 further regulates TWIK-1-mediated potassium currents in normal astrocytes. An RNA immunoprecipitation assay to determine how AEG-1 regulates the expression of TWIK-1 revealed that AEG-1 binds directly to TWIK-1 mRNA. Furthermore, TWIK-1 mRNA stability was significantly increased upon overexpression of AEG-1 in cultured astrocytes (*p* < 0.01). Our findings show that AEG-1 serves as an RNA-binding protein to regulate TWIK-1 expression in normal astrocytes.

## 1. Introduction

Astrocytes are the most well-studied glial cells in the brain, and recently, it has been reported that they are involved in pathological mechanisms of CNS diseases such as edema and neurodegeneration [1,2]. In particular, astrogliosis is a common feature of astrocytes that appear in pathological processes, and it is known that astrocyte-elevated gene-1 (AEG-1) increases during this process, and it is thus attracting attention as a novel treatment target [3]. AEG-1 was initially identified as an inducible gene in human fetal astrocytes infected with HIV-1 or treated with TNF-α [4]. Many studies of AEG-1 have reported to play a pivotal role in brain cancers as an oncogene, such as glioma and neuroblastoma, and in many other tissue cancers, but not in normal astrocytes [5,6,7,8,9,10]. AEG-1 is reportedly associated with tumorigenesis and serves as a multifunctional mediator involved in several signalling pathways including PI3K/Akt, NF-κB, Wnt/β-catenin, and MAPK [11,12,13]. Because AEG-1 does not contain any DNA-binding domains, it is believed to regulate these signal transduction pathways through interactions with several transcription factors. Indeed, AEG-1 interacts with transcription factors such as p65, c-Jun, Yin Yang 1, cyclic AMP responsive element-binding protein (CREB), and CREB-binding protein [14,15,16].

Recently, Hsu et al. reported that AEG-1 is an integral RNA-binding membrane protein in the endoplasmic reticulum (ER) that primarily binds to the coding sequence of ER mRNAs and transmembrane protein-encoding mRNAs [17,18]. Although AEG-1 lacks a canonical RNA-binding domain, it has been identified as an RNA-binding protein (RBP) by several AEG-1–RNA interactome screens [19,20,21]. However, its cellular function as an RBP is still unclear.

TWIK-1, a member of two-pore-domain potassium (K2P) channel family, is highly expressed in normal astrocytes [22]. We first reported that TWIK-1 is a heterodimer with TREK-1 in normal astrocytes, contributing to passive conductance and GPCR-mediated glutamate release [23,24]. Furthermore, we reported that TWIK-1 modulates intrinsic excitability by forming heterodimers with TASK-3 in the dentate gyrus granule cells (DGGC) neurons [25,26]. Despite the physiological significance of TWIK-1, few studies have focused on the molecular mechanisms regulating TWIK-1 expression [27].

In this study, we used shRNA to investigate AEG-1 function in normal astrocytes. This study shows that AEG-1 binds to TWIK-1 mRNA in the ER in astrocytes and regulates its expression.

## 2. Materials and Methods

### 2.1. Plasmids and shRNA

The cDNA encoding full-length of mouse AEG-1 (GenBank accession no. NM_026002) was synthesized by gBlocks (IDT, Singapore), and mouse TWIK-1 (GenBank accession no. NM_008430) was obtained using a RT-PCR-based Gateway (Invitrogen, Carlsbad, CA, USA). The constructs were subcloned into the pDEST-GFP-N vector by Gateway cloning. The shRNA vector against TWIK-1 or AEG-1 was described in our previous studies [24,28].

### 2.2. Primary Astrocyte Culture

Primary astrocyte cultures are prepared from the cerebral cortices of C57BL/6 mouse pups (P1) as described previously [29]. In brief, the cerebral cortex from 1-day-old C57BL/6 mice were chopped and mechanically disrupted through trituration. The cells obtained were seeded in culture flasks and grown at 37 ℃ in a 5% CO_2_ atmosphere in DMEM supplemented with 10% heat-inactivated FBS, 100 U/mL penicillin-streptomycin. The culture medium was changed after 3–4 days to eliminate debris and other floating cell types. Cells were used and transfected after 5–7 days of culture.

### 2.3. RNA Sequencing (RNA-Seq) and Analysis

Sequencing and analysis were performed by eBiogen (Seoul, Korea). Total RNA of primary cultured astrocytes was isolated as manufacturer (GeneAll, Seoul, Korea). Two samples were prepared for each group. Libraries were prepared from 2 μg of total RNA and isolated mRNA. The isolated mRNAs were used for cDNA synthesis and shearing. Indexing was performed using the Illumina index 1–12. Libraries were identified using the Agilent 2100 Bioanalyzer system to evaluate the mean fragment size. Quantification was performed using the library quantification kit using a StepOne RT-qPCR System. High-throughput sequencing was performed with a 100 bp paired-end protocol using HiSeq 2500 (Illumina, San Diego, CA, USA). mRNA-seq reads were mapped by TopHat software tool to obtain alignment files that were subsequently used to obtain assembling transcripts. mRNA transcript abundance and differential gene expression was analyzed using cufflinks (https://www.genecards.org). Quantile normalized FPKM by EdgeR in R was used to determine expression level of gene regions.

### 2.4. RT-PCR and qPCR

Total RNA of primary cultured astrocytes was isolated as manufacturer (GeneAll, Seoul, Korea). For RT-PCR and qPCR, 500 ng of total RNA was used for cDNA preparation with an SensiFAST cDNA Synthesis Kit (BIOLINE, London, UK) according to the manufacturer’s protocol. The following RT-PCR primers were used: TWIK-1, 5′-TGCTCTACCTGGTGTTCGG-3′ (forward) and 5′-GAGTAGATGATGCAGAAGGC-3′ (reverse); TREK-1, 5′ CAGAACTCCAAACCGAGGCT-3′ (forward) and 5′-GATGTTTCCAAATCCTATGG-3′ (reverse); Actin, 5′- CCCAGATCATGTTTGAGACC-3′ (forward) and 5′- TCATGGATGCCACAGGATTC-3′ (reverse); glyceride-3-phosphate dehydrogenase (GAPDH), 5′-GTCTTCACCACCATGGAGAA-3′ (forward) and 5′-GCATGGACTGTGGTCATGAG-3′ (reverse). PCR amplification was performed using the 2X TOPsimple DyeMIX-Tenuto (Enzynomics, Daejeon, Korea). qPCR was performed using the SensiFAST Probe Hi-ROX kit (BIOLINE). Primer sets for AEG-1 (Mm.PT.58.10641401), TWIK-1 (Mm.PT.58.11947413) and GAPDH (Mm.PT.39a.1) were purchased from IDT. GAPDH was used as an internal normalization control. Detailed sequence information for each is specified in Table 1. All experiments were performed in triplicates. The 2^−ΔΔCt^ method was applied to calculate fold changes in gene expression.

### 2.5. Immunocytochemistry

Primary cultured astrocytes were rinsed in PBS and fixed in 4% PFA for 15 min. Cells were permeabilised 0.1% Triton X-100 (Sigma-Aldrich, St. Louis, MO, USA) and blocked using 0.1% BSA for 1.5 h, then incubated with anti-TWIK-1 (Elpisbio, Daejeon, Korea), anti-AEG-1 (40-6500, Invitrogen, Carlsbad, CA, USA), and anti-GFAP (PA1-10004; Thermo Scientific, Waltham, MA, USA) antibodies overnight at 4 °C. This was followed by incubation for 1 h at room temperature with fluorescence-conjugated and mounted with DAPI-containing gelatin. Images were acquired by a Nikon A1 confocal microscope (Nikon Inc., Tokyo, Japan) at room temperature.

### 2.6. Electrophysiology

Every seventh day after transfection, cells were transferred to a perfusion bath with an artificial extracellular solution containing (in mM) 150 NaCl, 3 KCl, 2 CaCl2, 1 MgCl2, 10 HEPES, 5.5 D-glucose, and 20 sucrose (pH 7.4 was adjusted with NaOH). Glass pipette electrodes (PG150T-15, Harvard Apparatus, Kent, UK) of resistance 3–5 MΩ were filled with a pseudo-intracellular solution containing (in mM) 150 KCl, 1 CaCl2, 1 MgCl2, 5 EGTA, and 10 HEPES (pH 7.2 was adjusted with KOH). Cells were patched using patch clamp in whole cell mode, where current was injected for 1-s voltage ramps from +50 mV to −150 mV (from a holding potential of −60 mV) using Axopatch 700B amplifier. Data were acquired and digitized using Clampex 10.2 and DigiData 1550B, respectively (Molecular Devices, San Jose, CA, USA). Recordings were carried out at room temperature.

### 2.7. Western Blotting

Samples were lysed using lysis buffer (50 mM HEPES, 0.1% sodium deoxycholate, 1% Triton X-100, 1 mM PMSF and 0.1% SDS) with protease inhibitor cocktail (T&I, Chuncheon, Korea). The same amount of protein (40 μg/lane) was added to a polyacrylamide gel (10%) for each sample. Proteins separated by SDS-PAGE were electro-transferred to PVDF membranes, and the membranes were blocked using 5% non-fat milk. After blocking, membrane was incubated with primary antibodies. Antibodies were obtained from the following suppliers: anti-TWIK-1 (4D7, Santa Cruz Biotechnology, Santa Cruz, CA, USA); anti-AEG-1 (40-6500, Invitrogen, Carlsbad, CA, USA); and anti-β-actin (A5541, Sigma-Aldrich, St. Louis, MO, USA). Membrane was then washed and incubated with HRP-conjugated secondary antibodies and incubated at room temperature for 1 h. The blots were detected with ECL Western Blotting Substrate (Thermo Fisher Scientific, Waltham, MA, USA).

### 2.8. RNA Immunoprecipitation (RIP) Assay

RIP assays were conducted using the EZ-Magna RIP Kit (Merck Millipore, Billerica, MA, USA). After transfection with GFP-AEG-1, primary cultured astrocytes were suspended in RIP buffer with mouse IgG or anti-GFP (B-2, Santa Cruz Biotechnology, Dallas, TX, USA). GFP–AEG-1 was immunoprecipitated using protein A/G beads (Santa Cruz Biotechnology), and the amount of TWIK-1 or TREK-1 mRNA in the precipitates was determined through RT-PCR. To increase the sensitivity of the RT-PCR protocol, we performed a second amplification step (30 cycles for TWIK-1, 30 cycles for TREK-1, and 20 cycles for actin) using a 20-μL reaction mixture containing 1 μL of the first PCR amplification product (20 cycles). The final PCR products were confirmed through sequencing.

### 2.9. mRNA Stability Assays

Cultured astrocytes were treated with 5 µg/mL actinomycin D (Act D, Sigma-Aldrich, St. Louis, MO, USA) for different periods (0, 2, 4, 6, 8, 12, and 24 h). After extracting total RNA from Act D-treated cells, using the RiboX Reagent (GeneAll), RT-PCR was used to confirm the stability of TWIK-1 mRNA.

### 2.10. Statistical Analysis

Data are presented as mean ± SEM values. Between-group comparisons were performed using Student’s t-test, and multiple-group comparisons were performed using ANOVA followed by the Tukey’s test for post hoc analysis. For experiments with two independent variables, two-way ANOVA followed by the Tukey’s post hoc test was used. Statistical significance was considered at * *p* < 0.05, ** *p* < 0.01, and *** *p* < 0.001.

## 3. Results

### 3.1. AEG-1 Knockdown Down-Regulates TWIK-1 mRNA in Astrocytes

To investigate the RNA expression pattern of genes in the presence and absence of AEG-1, we performed RNA-Seq in scramble shRNA- (negative control) and AEG-1 shRNA-transfected astrocytes (AEG-1 knockdown) (Figure 1A). Bioinformatic analysis of RNA-Seq data revealed 132 up-regulated genes and 109 down-regulated genes in AEG-1 knockdown cells rather than in the negative control cells (Figure 1B, Appendix A). Further, Gene Ontology analysis revealed that most of the down-regulated genes were associated with ion transport pathways (Figure 1C). Thus, we examined the expression patterns of the genes in this GO category (ion transport) and those belonging to the GO category related to the ion channel complex (Figure 1D,E). The generated heat map representing the differences in gene expression indicated that TWIK-1 expression was positively associated with AEG-1 expression. Furthermore, TWIK-1 expression was significantly different in both ion transport and ion channel complex GO categories. qPCR analysis of TWIK-1 further confirmed that its expression was significantly reduced in AEG-1 shRNA-transfected astrocytes than in the negative control cells (*p* < 0.001) (Figure 1F).

### 3.2. AEG-1 Knockdown Down-Regulates TWIK-1 Protein and TWIK-1-Mediated Potassium Currents in Cultured Astrocytes

Since AEG-1 knockdown downregulates TWIK-1 mRNA, we investigated whether AEG-1 also regulates the amount and function of the TWIK-1 protein in cultured astrocytes. To determine the amount of endogenous TWIK-1 protein in accordance with AEG-1 expression levels, we performed immunocytochemistry for scramble shRNA- and AEG-1 shRNA-transfected astrocytes. First, we conducted a co-immunostaining experiment with antibodies against astrocyte marker GFAP and AEG-1 to confirm the purity of cultured astrocytes and expression of endogenous AEG-1, and most of the cells expressed both proteins simultaneously (Figure 2A). Figure 2B,C show that AEG-1 shRNA effectively reduced the levels of both proteins, AEG-1 and TWIK-1, compared with scramble shRNA-treated cells. The astrocyte has a very dynamic cytoskeleton, and if the amount of protein is large, there is a possibility that the cytoskeleton protein may be saturated [30,31]. So, we confirmed that there was no significant difference in the total amount of protein between samples by showing Ponceau S staining data. We previously reported that TWIK-1 mediates astrocyte potassium currents by forming a heterodimer with TREK-1, one of the other K2P subunits [21]. To assess the influence of AEG-1 on TWIK-1 function, we measured the whole-cell currents in differently treated astrocytes. Overexpression of AEG-1 and TWIK-1 shRNA, alone, and in combination, significantly reduced potassium currents in primary cultures of astrocytes (*p* < 0.01 and *p* < 0.001, respectively) (Figure 2D). The current density was then normalized at +50 mV and −150 mV compared to the scramble shRNA-transfected conditions (Figure 2E). These data indicate that suppression of AEG-1 decreases both the amount and function of TWIK-1 in cultured astrocytes.

### 3.3. AEG-1 Overexpression Up-Regulates TWIK-1 mRNA and Protein and Astrocytic Potassium Currents in Cultured Astrocytes

Considering the suppression of TWIK-1 expression by AEG-1 inhibition, we tested if TWIK-1 expression can be directly increased through AEG-1 overexpression. As expected, AEG-1 overexpression significantly up-regulated TWIK-1 mRNA and protein levels in cultured astrocytes (*p* < 0.001) (Figure 3A,B). Whole-cell current recordings revealed that the astrocytic potassium currents were also significantly increased upon AEG-1 overexpression (*p* < 0.05) (Figure 3C,D). These data show that AEG-1 overexpression enhances TWIK-1 expressions and astrocytic potassium currents in primary cultured astrocytes.

### 3.4. AEG-1 Is an RNA-Binding Protein That Enhances TWIK-1 mRNA Stability

We investigated how AEG-1 regulates TWIK-1 expression. We recently reported that AEG-1 potentially up-regulates another astrocytic K2P channel, namely, TREK-1 [25]. As a follow-up study, we performed a TREK-1 promoter assay to examine the regulatory mechanism; however, AEG-1 did not display any effects. We hypothesized that, since AEG-1 is an ER RBP [17], it may also bind to TWIK-1 mRNA and regulate its stability. Thus, we investigated whether AEG-1 can bind to TWIK-1 mRNA, through RNA immunoprecipitation, followed by RT-PCR (Figure 4A). As shown in Figure 4B, AEG-1 bound to endogenous TWIK-1 mRNA, but not to actin mRNA (control), in cultured astrocytes. We then tested the stability of TWIK-1 mRNA using the RNA polymerase inhibitor Act D, which is widely used to confirm the stability of existing mRNA by preventing new RNA synthesis [32]. Incubation of cultured astrocytes with Act D for 8 h significantly down-regulated TWIK-1 mRNA in comparison with the control (GAPDH) (Figure 4C,D). To determine the effect of AEG-1 on the stability of TWIK-1 mRNA, we performed qPCR in GFP- and GFP-AEG-1-transfected astrocytes after 8-h incubation with Act D. The stability of TWIK-1 mRNA was significantly increased in GFP-AEG-1-transfected astrocytes compared to that in the GFP-transfected sample (*p* < 0.01) (Figure 4E), indicating that AEG-1 acts as an RBP that regulates TWIK-1 mRNA stability.

## 4. Discussion

Most studies of AEG-1 have focused on its role in cancer and neuronal diseases. However, little is known about the function of AEG-1 in normal astrocytes. Recently, AEG-1 has been reported as an RBP that plays an essential role in the post-transcriptional gene expression [17]. RNA-binding proteins bind both RNAs and proteins and are thus involved in multiple biological processes, such as pre-mRNA splicing, modification, transport, transcription, translation, and stability [33,34,35].

Here, we unveiled that AEG-1, serves as an RNA-binding protein, regulates TWIK-1 expression in normal astrocytes. Furthermore, binding of AEG-1 to TWIK-1 mRNA prevented the mRNA from degradation in normal astrocytes, which led to the preservation of the function of TWIK-1 transcript and protein.

Since the expression of AEG-1 is up-regulated in cancer cells and increases with cancer progression, it is considered a therapeutic target for brain cancers. In contrast, the expression of AEG-1 is relatively low in healthy brains, where it is believed to exhibit neuroprotective effects. Indeed, in some degenerative neurological diseases, such as ALS and Parkinson’s disease, a decrease in neuronal AEG-1 has been reported. In addition, the up-regulation of AEG-1 led to neuronal protection in vivo [36,37].

We previously reported that AEG-1 regulates TREK-1, the latter of two pore potassium channels that are highly expressed in astrocytes. During acute hypoxia, hypoxia-inducing factor 1α (HIF-1α) levels are increased and directly induce AEG-1 expression, which in turn up-regulates TREK-1 in astrocytes [28]. The present results show that AEG-1 up-regulates TWIK-1 in astrocytes. Based on previous and current studies, astrocytic TWIK-1 and TREK-1 increased by AEG-1 potentially contribute to the protection of excitatory neuronal death by absorbing extracellular potassium ions secreted from activated neurons under pathological conditions. Thus, AEG-1 inhibition should be carefully reviewed as a therapeutic strategy for cancer treatment in future studies.

In this study, since gene profiling was analyzed using only 2D cultured astrocytes, there may be limitations in that there is variability in 3D cultured astrocytes and genetic and protein profiling. Future studies may utilize 3D self-constructing models and organ-on-a-chip platforms for advanced imaging that can apply spatial transcriptomes along with other imaging modalities such as extended microscopes to overcome previous limitations. Furthermore, the use of humanized models such as human iPSC-derived astrocytes and advanced spatial transcriptomic techniques such as MERFISH [38] are expected to provide more useful information on disease treatment.

## 5. Conclusions

In summary, we identified that AEG-1 has vital role to play in novel regulator of TWIK-1 in normal astrocytes. AEG-1 stimulates TWIK-1 expression and TWIK-1-mediated astrocytic potassium currents by stabilizing TWIK-1 mRNA. This represents a new AEG-1-mediated mechanism by which TWIK-1 is regulated under physiological or certain pathological conditions.

## Figures and Tables

**Figure 1 brainsci-11-00085-f001:**
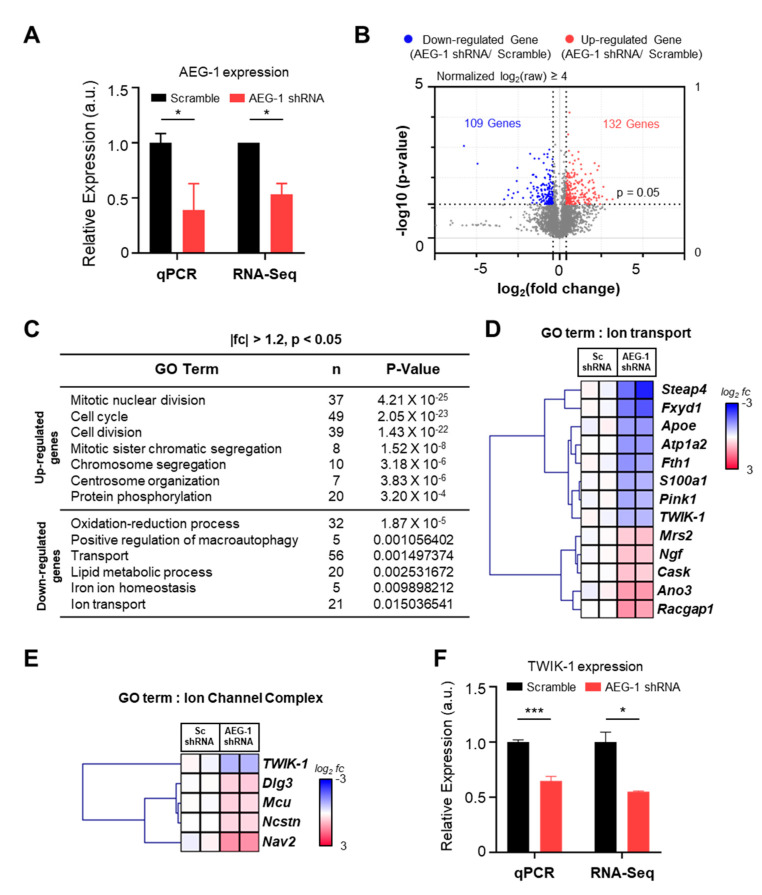
RNA-Seq and bioinformatic analysis of AEG-1 and its effect on TWIK-1 mRNA in cultured astrocytes. (**A**) Relative expression levels of AEG-1 mRNA in scramble (Sc) shRNA- and AEG-1 shRNA-transfected astrocytes. (**B**) Volcano plot of log_2_(fold change) for genes with values greater than 4 in scramble shRNA- and AEG-1 shRNA-transfected astrocytes. (**C**) GO Term analysis of differentially expressed genes. (**D**) Clustered heat map indicating the expression difference between genes in the ion transport GO category (red, up-regulated genes; blue, down-regulated genes). (**E**) Clustered heat map of genes in the ion channel complex GO category. (**F**) Relative expression levels of TWIK-1 mRNA in scramble (Sc) shRNA- and AEG-1 shRNA-transfected astrocytes. Data are presented as mean ± SEM. * *p*< 0.05, *** *p* < 0.001.

**Figure 2 brainsci-11-00085-f002:**
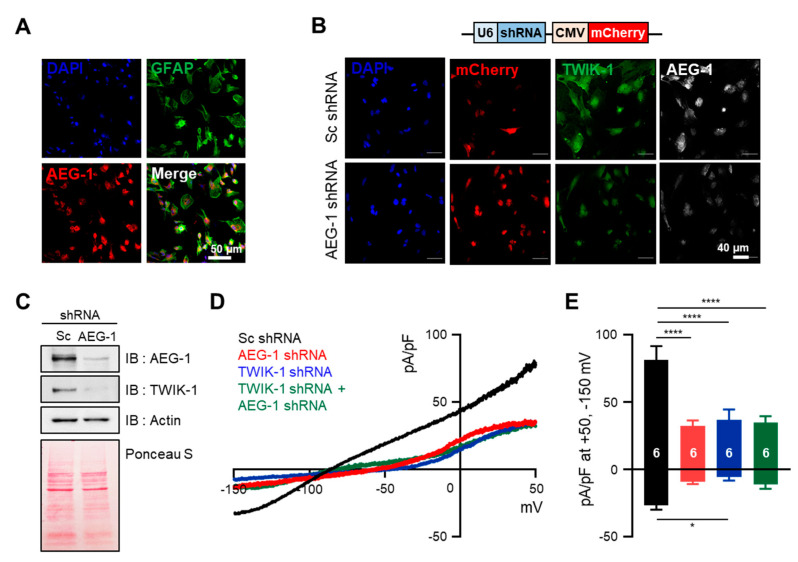
AEG-1 knockdown reduces TWIK-1 protein levels and TWIK-1-mediated potassium currents in primary cultured astrocytes. (**A**) Immunofluorescence staining of cultured astrocytes using AEG-1 and GFAP, an astrocyte marker. (**B**) Immunofluorescence staining of TWIK-1 and AEG-1 in transfected astrocyte. Knockdown of the AEG-1 inhibits the levels of TWIK-1. Scale bar, 40 μm. The shRNA vector contains a shRNA sequence under the control of U6 promoter, with mCherry co-expression under a CMV promoter. (**C**) Knockdown of the AEG-1 in primary cultured astrocytes. Immunoblot data show the decreased AEG-1 and TWIK-1 expression. (**D**) Representative whole-cell I–V curves of astrocytes overexpressing Sc shRNA (black), AEG-1 shRNA (red), TWIK-1 shRNA (blue), and AEG-1 and TWIK-1 shRNAs (green). (**E**) Pooled data for whole-cell current amplitudes in AEG-1 knockdown, TWIK-1 knockdown, and AEG-1 and TWIK-1 double knockdown astrocytes. All values are presented as mean ± SEM. * *p* < 0.05, **** *p* < 0.0001. Two-way ANOVA followed by the Tukey’s post hoc.

**Figure 3 brainsci-11-00085-f003:**
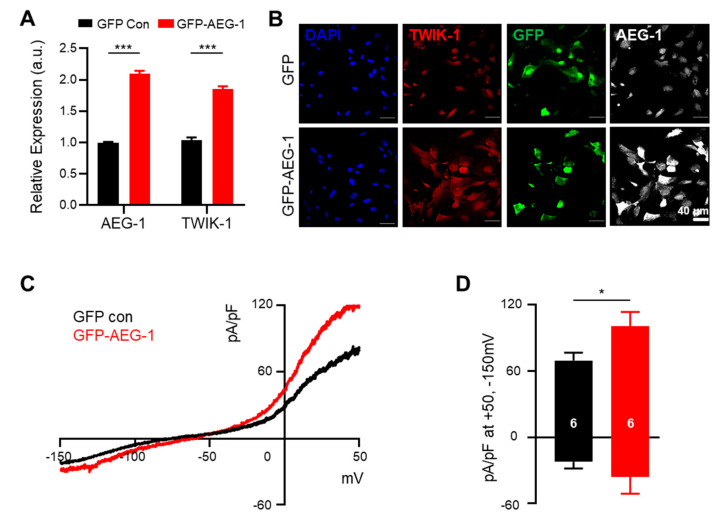
AEG-1 overexpression enhances TWIK-1 mRNA and protein levels and *astrocytic* potassium currents in cultured astrocytes. (**A**) Overexpression of AEG-1 increases TWIK-1 mRNAs (**B**) Immunofluorescence staining of TWIK-1 and AEG-1 in transfected astrocytes. AEG-1 overexpression increases expression levels of TWIK-1. Scale bar, 40 μm. (**C**) Representative traces from transfected astrocytes with GFP (control)- and GFP-AEG-1. (**D**) Bar graph showing potassium currents averaged from results in (**C**) at +50 mV and −150 mV. The number on each bar indicates n for each condition. All values are presented as mean ± SEM. * *p* < 0.05, *** *p* < 0.001. P-values were obtained from the Student’s t-test.

**Figure 4 brainsci-11-00085-f004:**
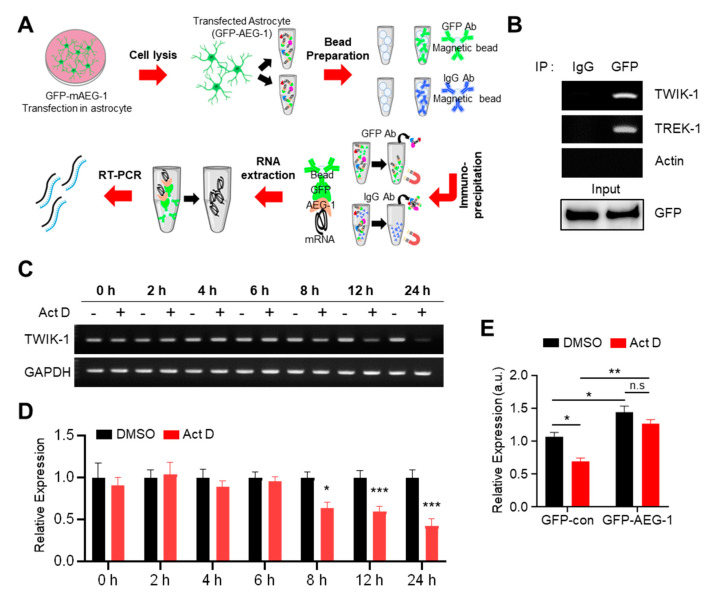
AEG-1 binds with TWIK-1 mRNA and enhances its mRNA stability. (**A**) Scheme of the RIP assay. (**B**) Immunoprecipitation of TWIK-1, TREK-1, and actin (negative control). RIP assay was performed on transfected astrocytes with GFP-AEG-1. Immunoprecipitation was performed with mouse IgG and GFP antibodies. Total RNA was purified, and TWIK-1 and TREK-1 mRNAs were detected by RT–PCR. Five percent of the total lysate was used as an input control. Actin was used as the negative control. Primary cultured astrocytes were incubated with Act D (5 μg/mL), and total RNA was harvested after various incubation times. TWIK-1 mRNA levels were determined using RT-PCR (**C**) and normalized to the GAPDH (**D**). Data show the mean ± SEM. * *p* < 0.05, *** *p* < 0.001 (Student’s t-test). (**E**) Relative expression levels of TWIK-1 mRNA after additional 8 h incubation with Act D (5 µg/mL) in transfected astrocytes with GFP-AEG-1. Total RNA was harvested from astrocytes after eight hours of incubation. Data are presented as mean ± SEM. * *p* < 0.05, ** *p* < 0.01. P-values were obtained from the two-way ANOVA with Tukey’s multiple comparison post hoc test.

**Table 1 brainsci-11-00085-t001:** qRT-PCR Primers and Probes.

Gene	Sequence
*AEG-1*	
Forward Primer	5′-CTATCTTCATCTACCCAGTTCCC-3′
Probe	5′-/56-FAM/CCTAGCTCA/ZEN/GACTGGAATGCACCA/31ABkFQ/-3′
Reverse Primer	5′-GGATGTTAGCCGTAATCAACCT-3′
*TWIK-1*	
Forward Primer	5′-GCTACAACCAGAAGTTCCGA-3′
Probe	5′-/56-FAM/CCGAGGAGC/ZEN/AGGTAACACGTGAT/3IABkFQ/-3′
Reverse Primer	5′- TTCAGCTCGTGGAGTTCAC -3′
*GAPDH*	
Forward Primer	5′-GTGGAGTCATACTGGAACATGTAG-3′
Probe	5′-/56-FAM/TGCAAATGG/ZEN/CAGCCCTGGTG/3IABkFQ/-3′
Reverse Primer	5′-AATGGTGAAGGTCGGTGTG-3′

## Data Availability

The data presented in this study are available on request from the corresponding author.

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
