# Peer review of "AEG-1 Regulates TWIK-1 Expression as an RNA-Binding Protein in Astrocytes"

_brainsci, 2021, doi:10.3390/brainsci11010085_

Round 1

Reviewer 1 Report

-Authors need to start the introduction with something more general about the increasing interesting in astrocytes for CNS disorders. Targeting the mechanistic pathway in astrocytes have recently been shown to provide a viable therapeutic option for some diseases such as CNS edema and neurodegeneration where there’s a clear unmet clinical need.

-Authors discussed the identification of three suggested pathways. What was the platforms for data analysis and has KEGG been considered for data analysis? Please refer to similar studies where KEGG was performed since it has the most extensive molecular pathways database:

- https://pubmed.ncbi.nlm.nih.gov/33126586/

- https://www.ncbi.nlm.nih.gov/pmc/articles/PMC5735114/

-The study is based on transcriptomic profiling of cells which are grown in 2D. New lines of research have discussed the variability of genetic and protein profiling in 2D vs 3D. Authors need to discuss this as a limitation of the current study. Future studies can benefit from 3D self-organized models and organ-on-a-chip platforms for advanced imaging where spatial transcriptomics can be applied in combination with other imaging modalities such as expansion microscopy.

-Authors need to indicate the above-mentioned limitations towards the end of the discussion and point out to future directions such as the humanized model and the use of advanced spatial transcriptomic techniques such as MERFISH.

Reviewer 2 Report

In their paper entitled “AEG-1 regulates TWIK-1 expression as an RNA-2 binding protein in astrocytes” the Authors report that the protein encoded by Astrocyte elevated gene-1 (AEG-1) is an RNA-binding protein able to regulate stability of the mRNA for TWIK-1, a protein that controls potassium currents in astrocytes.

The paper is suitable for Brain Sciences, the methods used are clearly described, and the results are of interest.

However, a couple of points should be considered before acceptance:

  1. The paper does not contain any demonstration that the cells in culture are indeed astrocytes (it might be useful, for example, to add a picture showing that the cells are GFAP-positive); by the way, on page 3 (lines 119-120) a sentence refers to immunohistochemistry with anti-GFAP antibodies, but this sentence has been deleted;
  2. On page 4, line 140, the Authors report that they loaded 40 ug of proteins per lane of the gels that were then used for the western blot analysis; they also report (see Figure 2) that they normalized their western analysis respect to beta-actin; this fact might represent a source of error (see, for example, Taylor and Posch, Biomed Res Int 2014;2014:361590. doi: 10.1155/2014/361590; and Pillai-Kastoori L et al., Anal Biochem. 2020, 593:113608. doi: 10.1016/j.ab.2020.113608): cytoskeletal proteins are indeed highly represented in cell extracts (especially when we consider that high amounts of total proteins were loaded on the gels), and the western blots for these proteins can go to saturation and thus to non-linear results; moreover, and in spite of the previous comment, in figure 2, the actin signal appears to be slightly less intense in the extracts from cells in which the expression of AEG-1 was suppressed. In addition, astrocytes have a dynamic cytoskeleton and the production of cytoskeletal proteins also depends on culture conditions. This point should be at least commented in the text.

Author Response

In their paper entitled “AEG-1 regulates TWIK-1 expression as an RNA-2 binding protein in astrocytes” the Authors report that the protein encoded by Astrocyte elevated gene-1 (AEG-1) is an RNA-binding protein able to regulate stability of the mRNA for TWIK-1, a protein that controls potassium currents in astrocytes.

The paper is suitable for Brain Sciences, the methods used are clearly described, and the results are of interest.

However, a couple of points should be considered before acceptance:

  1. The paper does not contain any demonstration that the cells in culture are indeed astrocytes (it might be useful, for example, to add a picture showing that the cells are GFAP-positive); by the way, on page 3 (lines 119-120) a sentence refers to immunohistochemistry with anti-GFAP antibodies, but this sentence has been deleted;

Thanks for your comment. We added immunostaining data with GFAP antibody in Figure 2A to show the purity of cultured astrocytes (line 191-194).

  1. On page 4, line 140, the Authors report that they loaded 40 ug of proteins per lane of the gels that were then used for the western blot analysis; they also report (see Figure 2) that they normalized their western analysis respect to beta-actin; this fact might represent a source of error (see, for example, Taylor and Posch, Biomed Res Int 2014;2014:361590. doi: 10.1155/2014/361590; and Pillai-Kastoori L et al., Anal Biochem. 2020, 593:113608. doi: 10.1016/j.ab.2020.113608): cytoskeletal proteins are indeed highly represented in cell extracts (especially when we consider that high amounts of total proteins were loaded on the gels), and the western blots for these proteins can go to saturation and thus to non-linear results; moreover, and in spite of the previous comment, in figure 2, the actin signal appears to be slightly less intense in the extracts from cells in which the expression of AEG-1 was suppressed. In addition, astrocytes have a dynamic cytoskeleton and the production of cytoskeletal proteins also depends on culture conditions. This point should be at least commented in the text.

I agree with your advice and concerns. So, we added the Ponceau S staining data, which can show the amount of total protein, to Figure 2C, and also included the above reference and explanation in the text (line 196-199).

This manuscript is a resubmission of an earlier submission. The following is a list of the peer review reports and author responses from that submission.

Round 1

Reviewer 1 Report

The study by Jung and Kim et al reports a novel regulatory role of AEG-1 on TWIK1 in normal astrocytes through stimulating its expression and TWIK1 mediated astrocytic currents by stabilizing TWIK1 at genetic level. This study is based on previous studies by the same group and it can open the doors to investigate the potential therapeutic role of inhibiting AEG1 in cancer.

The design of the study and the technical quality of the work are convincing and results can be of general interest. The manuscript is well-written and easy to follow. Authors used correct statistical approaches in analyzing the results and the data is well-presented.

Reviewer 2 Report

In general

The text is very unprecise and there are inconsistencies between the figures, the M&M section and the results. There is no quantification of the results in transfected cells.  The number of replicates is never given. How are done statistics in this study?  There is no official deposit of the RNAseq raw data. All transfections are done at DIV5. At this stage, these cells are not yet differentiated astrocytes. If a GFAP staining was done, it would be completely negative. If a Dapi nuclear staining was given it would probably show that the cells that are used here are a mix of neural cells. There is finally no way to conclude that AEG-1 directly binds TWIK-1 mRNA.  To give this conclusion, it should be tested. Other partners might be implicated.

Figure 1:

RNAseq: A supplementary table with the 99 downregulated genes and the 173 upregulated genes is missing. It should be given with the fold change the number of reads and the p value. How many independent libraries have been analyzed. How was performed the alignment? What are the statistical tests to compare scramble and shRNA experiments?

Figure 1C: Is the table only for downregulated genes? How are the GO terms picked and how are they classified in the table? What does represent the line in the middle of the table? Explain why the downregulation FC threshold is 1.2, and 2 for upregulation.

Figure1D:  legends say ‘blue, up-regulated genes; red, down-regulated genes. It is the opposite.

Figure 1E: The same scale than in  1D should be given. Why blue intensity is different for TWIK-1 in Fig. 1D and 1E.

Figure 2

Figure 2A: The legend is not clear.  There is no explanation anywhere about this mCherry signal. These images are not at all convincing. Image of untransfected astrocytes should be given. A nuclear staining should be added to see all cells as well as a GFAP labelling to see which cells are astrocytes. A Western blot (WB) analysis should be done in parallel to show the decrease of AEG1 and TWIK1.

Figure 2B: A new shRNA TWIK-shRNA is used here. qPCR, immunofluorescence and WB tests should be done to characterized it.   

Figures 2D and E are the same than 2C.

Figure 3

Legends do not describe the figure. Quantifications are required to compare transfected and untransfected cells. Here, the whole potassium current is recorded and not the TWIK-mediated potassium current.

Figure 4

Figure and text show GFP-AEG-1 transfection while material and methods talks about GFP-TWIK-1 transfection….Why not to use directly an IP against endogenous AEG1 to analyze mRNAs linked to AEG1? Transfected cells overexpress AEG1 and this might create artefacts. Redo the experiments with an AEG1 antibodies or explain why it is not possible?

Figure 4C: Quantification of the effect is needed.

Others

The ref is missing in line 53 ‘few studies have focused on the molecular mechanisms regulating TWIK-1 expression’.

In line 215, maybe it is ‘translation’ instead of ‘transcription’?

Figure 1A and 1F replace pRCR by qPCR